# Induced Pluripotent Stem Cell-Based Cancer Immunotherapy: Strategies and Perspectives

**DOI:** 10.3390/biomedicines13082012

**Published:** 2025-08-19

**Authors:** Xiaodong Xun, Jialing Hao, Qian Cheng, Pengji Gao

**Affiliations:** 1Department of General Surgery, Beijing Jishuitan Hospital, Capital Medical University, Beijing 100035, China; xunxiaodong@jst-hosp.com.cn; 2Beijing Research Institute of Traumatology and Orthopedics, Beijing 100035, China; 3Department of Hepatobiliary Surgery, Peking University People’s Hospital, Beijing 100044, China; 2311110397@stu.pku.edu.cn; 4Peking University Institute of Organ Transplantation, Peking University, Beijing 100044, China

**Keywords:** induced pluripotent stem cells, immunocytes, cancer, cellular immunotherapy

## Abstract

Cellular immunotherapy has emerged as a transformative approach in oncology, revolutionizing cancer treatment paradigms. Since the groundbreaking development of induced pluripotent stem cells (iPSCs) by Yamanaka in 2008, significant progress has been made in generating various iPSCs-derived immunocytes, including T cells, dendritic cells, macrophages, natural killer (NK) cells, and B cells. These engineered immune cells offer unprecedented opportunities for personalized cancer therapy as they can be derived from patients’ own cells to minimize immune rejection. In addition, various new techniques are being used for the induction and amplification of iPSCs-derived immunocytes, such as small-molecule techniques, 3D culture systems, nanotechnology, and animal models for the in vivo amplification of immunocytes. Of course, challenges remain in improving immunocyte characteristics. Targeting efficiency needs enhancement to better distinguish tumor cells from healthy tissue, while biological activity must be optimized for sustained antitumor effects. Safety concerns, particularly regarding potential off-target effects and cytokine release syndrome, require further investigation. The immunosuppressive nature of tumor microenvironment also poses significant hurdles for solid tumor treatment. Ongoing clinical trials are exploring the therapeutic potential of iPSCs-derived immunocytes, with researchers investigating combination therapies and genetic modifications to overcome current limitations.

## 1. Introduction

Currently, surgical intervention is one of the first-line treatments for patients with solid tumors [1]. Most postoperative patients receive chemotherapy to prevent tumor relapse; however, most chemotherapy agents gradually lose effectiveness in these patients after several treatment cycles, leading to eventual recurrence of the malignancy. Furthermore, systemic treatment options for patients with hepatocellular carcinoma are limited to sorafenib and other targeted therapies (e.g., lenvatinib, regorafenib, cabozantinib, and ramucirumab) [2]. At this time, immunotherapy may be one of the most appropriate treatment options to prevent tumor recurrence in these patients. The currently used novel approach of cancer immunotherapy consists of novel monoclonal antibodies, novel immune adjuvants, and immune-modulating antibodies used for targeting regulatory cells, therapeutic cancer vaccines, and adoptive T cell therapies (Figure 1) [3]. Immunocytes play vital roles in these novel immunotherapy approaches. Monoclonal antibodies facilitate the exposure of tumor-associated antigens to dendritic cells (DCs), which present the antigens to T cells [4]. Immune-modulating antibodies, such as PD-1-blocking antibodies, are capable of augmenting the responses of T cells in vitro [5]. DC vaccines are important therapeutic cancer vaccines, and some clinical trials have been conducted in cancer patients with encouraging preliminary results [6]. Adoptive T cell therapy relies on the expansion of cancer-reactive T cells that are harvested from cancer patients and reintroduced into them to generate antitumor immunity [7].

As explained, immunocytes are among the most important and central components of most immunotherapies. However, there are some drawbacks to using immunotherapies with patient-derived primary immunocytes [8]. The primary immunocytes collected from patients are easily exhausted during expansion culturing. Additionally, it is costly to culture immunocytes, and in some cases, it is difficult to obtain potent immunocytes from patients. The generation of induced pluripotent stem cells (iPSCs) resolves these problems. iPSCs can address these issues by providing a consistent and scalable source of immune cells, including T cells, natural killer (NK) cells, and dendritic cells, which can be engineered to target specific tumor antigens effectively [9]. Additionally, iPSCs-derived immune cells exhibit enhanced functionality, including improved cytotoxicity and persistence compared to their primary counterparts, making them a compelling option for adoptive cell therapies [10].

Currently, an increasing number of groups have attempted to generate immunocytes with iPSCs for cell-based cancer immunotherapy. Here, in the following sections, we review the conventional protocols for the production of iPSCs-derived immunocytes and some new strategies that may improve various aspects of their production. Although the benefits of antitumor immunotherapy have not been demonstrated in a wide range of tumor types, cancer vaccine strategies have been proven to be a critical element of the treatment options for cancer patients [11]. iPSCs have not been approved for use in the clinic; nevertheless, the current cell-based immunotherapy with autologous immunocytes lays a foundation for future clinical applications with iPSCs.

## 2. The Basic Characteristics of iPSCs and Their Integration with Cancer Immunotherapy

### 2.1. The Sources and Characteristics of iPSCs

A tremendous breakthrough in the generation of stem cells was achieved in 2006, when Professor Yamanaka’s group demonstrated that pluripotent stem cells can be produced by transferring four transcription factors into mouse fibroblast cells [12]. These four transcription factors, which are important for the functions of embryonic cells, are Oct-3/4, Sox2, c-Myc, and Klf4. These cells are called induced pluripotent stem cells (iPSCs) and have similar genes and functions as embryonic stem cells (ESCs), including similar patterns of genetic and epigenetic expression, cell division rates, the ability to form embryoid bodies, induce tumorigenicity, and generate chimeras. Thus, iPSCs have similar characteristics to ESCs. iPSCs also have some advantages over ESCs, which will make them convenient to use in future applications in the clinic [13]. The generation of ESCs requires oocytes and the destruction of embryos, which raises ethical issues. Moreover, the immunogenicity of ESCs and recipients is different; for example, the transfusion of ESCs can potentially induce immunological rejection. These drawbacks of ESCs can be overcome by using iPSCs because the iPSCs used for therapy are acquired from autologous tissues; that is, ethical issues and immunological rejection are not concerns when iPSCs are used. However, concerns regarding the safety and potential tumorigenicity of iPSCs remain, necessitating rigorous evaluation and optimization of differentiation protocols to ensure the generation of safe and effective therapeutic cells (Table 1).

Intracellular signaling pathways also play a crucial role in somatic cell reprogramming. Various signaling pathways, including the Wnt, MAPK, and PI3K/Akt pathways, are known to influence the reprogramming process. Understanding how these signaling pathways interact with transcription factors and epigenetic modifications can provide insights into optimizing reprogramming protocols and improving the quality of iPSCs [14].

Epigenetic modifications are critical for the successful reprogramming of somatic cells to iPSCs. These modifications, including DNA methylation and histone modifications, regulate gene expression without altering the underlying DNA sequence. During the reprogramming process, there is a dynamic restructuring of the epigenome, which facilitates the transition from a differentiated state to a pluripotent state. Thus, targeting epigenetic regulators presents a promising strategy for improving reprogramming outcomes and generating high-quality iPSCs [15].

### 2.2. Types and Applications of Reprogramming Vectors

Reprogramming vectors are essential tools in the field of regenerative medicine, enabling the conversion of somatic cells into induced pluripotent stem cells (iPSCs) or other cell types. The choice of vector significantly influences the efficiency, safety, and applicability of the reprogramming process. Broadly, these vectors can be classified into two categories: viral vectors and non-viral vectors, each with distinct advantages and limitations.

Viral vectors, specifically those originating from retroviruses, lentiviruses, and adenoviruses, are extensively employed for gene delivery owing to their high transduction efficiency and capability to integrate into the host genome. A significant advantage of these viral vectors lies in their ability to convey substantial genetic payloads, which is essential for the simultaneous introduction of multiple reprogramming factors [16]. Nonetheless, the use of viral vectors is not without considerable drawbacks. The potential for immune responses directed against viral proteins can result in the elimination of transduced cells, and there are apprehensions regarding the long-term stability of transgene expression. Additionally, the production process for viral vectors can be intricate and expensive, presenting challenges for large-scale implementations [17]. These considerations underscore the necessity for meticulous evaluation when choosing viral vectors for reprogramming purposes.

In contrast, non-viral vectors have surfaced as viable alternatives to viral systems, presenting several advantages such as reduced immunogenicity, simplified production processes, and enhanced safety for applications. Techniques such as electroporation, lipofection, and nanoparticle utilization have been investigated to improve the delivery of reprogramming factors while minimizing the risks associated with viral integration [18]. Moreover, the use of CRISPR/Cas9 technology in conjunction with non-viral vectors has opened new avenues for precise genome editing, further enhancing the potential of these systems in regenerative medicine [19]. Despite these advancements, challenges remain, particularly in optimizing the efficiency of non-viral delivery methods and ensuring robust expression of transgenes (Table 2).

### 2.3. iPSCs Reprogramming Standard Protocol

Induced pluripotent stem cells (iPSCs) have emerged as a revolutionary tool in regenerative medicine, allowing for the generation of pluripotent cells from somatic cells. The reprogramming of somatic cells to iPSCs involves a series of meticulously defined protocols that ensure the successful conversion of differentiated cells back to a pluripotent state. This section outlines the standard protocol for iPSCs reprogramming, focusing on the cell source, culture conditions, detailed steps of the reprogramming process, and strategies for optimizing success rates.

The choice of cell source is critical for the efficient generation of iPSCs. Commonly used somatic cells include fibroblasts, peripheral blood mononuclear cells (PBMCs), and keratinocytes. Each cell type offers distinct advantages in terms of accessibility, reprogramming efficiency, and ethical considerations. The culture conditions also play a vital role in the reprogramming process. Cells are typically cultured in a defined medium that supports their growth and maintains their undifferentiated state. The use of feeder layers or extracellular matrix components can enhance cell attachment and proliferation. Furthermore, the environmental conditions, such as oxygen levels and temperature, should be optimized to mimic physiological conditions.

The reprogramming process generally follows a series of well-defined steps. Initially, somatic cells are harvested and cultured under optimal conditions. Following the introduction of reprogramming factors, cells are monitored for morphological changes indicative of pluripotency, such as colony formation and the expression of pluripotency markers (e.g., Nanog, SSEA-4). The reprogramming process typically spans several weeks, during which cells are subjected to selective culture conditions that favor the survival of successfully reprogrammed iPSCs. Once colonies are established, they can be expanded and characterized for pluripotency using assays such as teratoma formation and differentiation into all three germ layers [16].

The success rates of iPSCs generation can vary significantly based on multiple factors, including the cell source, reprogramming method, and culture conditions. Generally, the efficiency of reprogramming can range from 0.1% to several percent, depending on the methodology employed [20]. To optimize success rates, several strategies can be implemented. One effective approach is the use of small molecules that enhance reprogramming efficiency by modulating cellular signaling pathways. Additionally, optimizing the timing and dosage of reprogramming factors can lead to improved outcomes. Another critical aspect is the monitoring of cellular health throughout the reprogramming process. Regular assessments of cell viability, proliferation rates, and the expression of pluripotency markers can help identify optimal conditions and adjust protocols accordingly. Furthermore, the integration of advanced techniques such as single-cell RNA sequencing can provide insights into the dynamics of reprogramming and facilitate the identification of factors that enhance efficiency [16].

### 2.4. Quality Control Parameters for Functional iPSCs-Derived Immune Cells

Quality control in the production of functional induced pluripotent stem cells (iPSCs)-derived immune cells is crucial to ensure their efficacy and safety for therapeutic applications. This involves a series of assessments that evaluate the cells’ phenotypic characteristics, functional capabilities, proliferation rates, and immune reactivity. Each parameter plays a significant role in determining the overall quality of the derived immune cells, ultimately impacting their potential use in clinical settings.

The assessment of cell phenotype and function is foundational in the quality control of iPSCs-derived immune cells. This process typically involves the analysis of surface markers and functional assays to confirm that the cells exhibit the desired characteristics of specific immune cell types. Furthermore, functional assays, such as cytokine production and cytotoxicity tests, are conducted to ascertain the cells’ ability to respond to stimuli and perform their intended immune functions.

Monitoring cell proliferation and viability is another essential quality control parameter for iPSCs-derived immune cells. This is typically achieved through assays that measure cell growth rates and survival under various conditions. Techniques such as the MTT assay, trypan blue exclusion, and live/dead staining are commonly used to assess the viability of the cells. Additionally, proliferation can be quantified using methods like the BrdU incorporation assay or CFSE dilution assays, which help in determining the growth characteristics of the cells over time. Maintaining optimal proliferation rates is vital, as it ensures a sufficient yield of immune cells for therapeutic applications while preventing issues related to senescence or loss of pluripotency [15].

Immunoreactivity testing is crucial for evaluating the functionality of iPSCs-derived immune cells, particularly in their ability to mount an immune response. This involves assessing the cells’ responsiveness to antigens or mitogens, which can be measured through various immunological assays. For example, enzyme-linked immunosorbent assays (ELISA) can be used to quantify cytokine production in response to specific stimuli, while mixed lymphocyte reactions (MLR) can evaluate the cells’ ability to proliferate and activate in the presence of foreign antigens. The assessment of immunoreactivity not only provides insights into the cells’ functional capabilities but also helps identify any potential deviations from expected immune profiles, which is essential for ensuring the safety and efficacy of these cells in therapeutic contexts [21].

### 2.5. The Differentiation Process of Immune Cells Derived from iPSCs and Their Advantages and Disadvantages

The differentiation of immune cells from induced pluripotent stem cells (iPSCs) is a promising area of research that holds significant potential for therapeutic applications in regenerative medicine and immunotherapy. iPSCs can be generated from somatic cells through reprogramming, allowing for the creation of patient-specific immune cells. The differentiation process involves directing iPSCs to develop into various immune cell types, such as T cells, B cells, and natural killer (NK) cells, through specific culture conditions and signaling pathways. This process typically includes the use of growth factors and cytokines that promote lineage specification and maturation. Understanding the pathways and factors involved in this differentiation process is essential for optimizing protocols to generate functional immune cells that can effectively respond to pathogens or tumors.

The differentiation pathways of immune cells derived from iPSCs are complex and involve multiple stages that mimic natural hematopoiesis. Initially, iPSCs are induced to form hematopoietic progenitor cells (HPCs), which can then be further directed to differentiate into specific immune cell lineages. Each immune cell type has a unique differentiation protocol that must be carefully optimized to ensure the production of cells with the desired functional characteristics. Furthermore, the application of genetic engineering techniques, such as CRISPR/Cas9, can enhance the specificity and efficacy of the differentiated immune cells, allowing for the development of engineered T cells or NK cells that target specific tumor antigens [22].

One of the main advantages of using iPSCs-derived immune cells is the potential for generating patient-specific therapies that minimize the risk of immune rejection. Since iPSCs can be derived from the patient’s own cells, the resulting immune cells are less likely to provoke an immune response when reintroduced into the patient. Additionally, iPSCs provide an unlimited source of cells for research and therapeutic applications, overcoming the limitations associated with primary immune cells, which are often difficult to obtain and expand in culture. iPSCs-derived immune cells also exhibit enhanced proliferative capacity and can be genetically modified to express chimeric antigen receptors (CARs) or other therapeutic molecules, increasing their effectiveness in targeting tumors. Furthermore, the ability to produce large quantities of uniform immune cells allows for standardized manufacturing processes, which is crucial for clinical applications. These advantages position iPSCs-derived immune cells as a promising tool for developing advanced immunotherapies for various diseases, including cancer and autoimmune disorders.

Despite the numerous advantages, the use of iPSCs-derived immune cells also presents several challenges and potential defects. One significant concern is the risk of tumorigenicity associated with iPSCs, as they retain the ability to form teratomas if not fully differentiated [23]. Additionally, there is a possibility that iPSCs-derived immune cells may exhibit altered functionality compared to their primary counterparts, potentially impacting their effectiveness in therapeutic applications [24]. Another challenge is the complexity of the differentiation protocols, which can lead to variability in the quality and functionality of the derived immune cells. Furthermore, the immunogenicity of iPSCs-derived cells, particularly in allogeneic settings, remains a concern, as they may still elicit immune responses despite being derived from the same patient [25]. Addressing these challenges will be crucial for the successful translation of iPSCs-derived immune cells into clinical practice, necessitating ongoing research to refine differentiation protocols and enhance the safety and efficacy of these cells (Table 3).

## 3. Differentiating iPSCs into Immunocytes: A Potential Cellular Source for Cancer Immunotherapy

### 3.1. iPSCs-Natural Killer Cells (NK Cells)

Peripheral blood mononuclear cells (PBMCs) are NK cells that were first discovered in the 1970s [30]. Currently, there are several groups generating autologous NK cells from the peripheral blood of humans [31]. NK cells can also be isolated from bone marrow-derived mononuclear cells [32]. However, due to the complexity of the procedure and the morbidity associated with aspiration of bone marrow, peripheral blood is preferred to bone marrow blood for the isolation of NK cells. Recently, some strategies based on the use of allogeneic NK cells instead of autologous NK cells were proposed for some specific applications to overcome these drawbacks [33].

Autologous natural killer (NK) cells are a promising component of immunotherapy, particularly in the context of cancer treatment. These cells are derived from the patient’s own immune system, which offers several advantages, including reduced risk of immune rejection and enhanced compatibility with the patient’s biological environment. Recent studies have demonstrated the feasibility of using autologous NK cells in clinical settings, particularly for patients with hematological malignancies, where they have shown promising results in reducing minimal residual disease post-transplantation [34]. Furthermore, the use of autologous cells allows for a personalized treatment approach, where the patient’s unique immune profile can be leveraged to optimize therapeutic outcomes. This personalized aspect is particularly vital in cancer therapy, where the immune landscape can significantly influence treatment success [35]. Despite the advantages of autologous NK cells, there are functional issues that can impact their efficacy in therapeutic applications. One of the primary concerns is the functional exhaustion of NK cells, which can occur due to the immunosuppressive environment often present in cancer patients. Factors such as chronic exposure to tumor-derived cytokines can lead to a state of reduced cytotoxicity and impaired cytokine production in NK cells [36]. Additionally, the activation and education of NK cells through interactions with other immune cells and the tumor microenvironment are crucial for their functionality. Studies have shown that the presence of inhibitory receptors, such as killer immunoglobulin-like receptors (KIRs), can negatively influence NK cell activity, particularly in the context of tumor recognition and elimination [25]. Allogeneic natural killer (NK) cells are a promising area of research in immunotherapy, particularly due to their unique characteristics that differentiate them from other immune cells. Unlike T cells, NK cells do not require prior sensitization to recognize and kill target cells, allowing them to act swiftly against tumors and virally infected cells. This innate ability is attributed to their recognition of stress signals and the absence of major histocompatibility complex (MHC) class I molecules on target cells. The allogeneic nature of NK cells, sourced from healthy donors, provides an “off-the-shelf” therapeutic option that can be readily available for patients without the need for extensive matching, which is often required for T cell therapies. This characteristic is particularly beneficial in treating hematological malignancies and solid tumors where rapid intervention is critical [37]. Furthermore, the ability of NK cells to exert a graft-versus-tumor effect without causing significant graft-versus-host disease (GvHD) makes them an attractive option for allogeneic transplantation settings [38]. While the use of allogeneic NK cells offers many advantages, there are inherent risks associated with immunological mismatches. The primary concern is the potential for graft-versus-host disease (GvHD), where the infused NK cells may attack the recipient’s healthy tissues, particularly in cases where there is significant HLA mismatch between donor and recipient [38]. Additionally, the activation of allogeneic NK cells can lead to the production of proinflammatory cytokines, which may exacerbate inflammation and contribute to adverse effects in the recipient [37]. Furthermore, the risk of developing donor-specific antibodies (DSAs) can complicate the clinical outcomes, as these antibodies may lead to acute rejection of the graft or chronic allograft dysfunction [39]. Therefore, careful donor selection and pre-transplant immunological assessments are crucial to mitigate these risks and enhance the safety and efficacy of allogeneic NK cell therapies.

ESCs and iPSCs are important cell sources for developing immunocytes that have the potential to treat both malignant and nonmalignant diseases. Several groups have demonstrated that hematopoietic progenitor cells derived from ESCs or iPSCs can differentiate into functional NK cells, which can serve as “unlimited” anticancer lymphocytes for cancer immunotherapy [40]. The initial protocol based on culturing ESCs in an OP9 coculture system yields as many as 20% CD34+ hematopoietic precursors [41]. It has been reported that more CD34+ CD45+ cells are derived from ESCs and that these types of hematopoietic precursors are more suitable than CD34+ CD45− cells for differentiation into NK cells [42]. Other protocols for the generation of CD34+ hematopoietic precursors in vitro are based on the formation of the embryoid body (EB), which is used in the OP9 coculture system with a cocktail of cytokines that also includes BMP4, VEGF, SCF, FGF, TPO, and Flt3L [43,44]. Knorr et al. [45] induced the formation of spin EBs in the presence of BMP-4 and VEGF and added IL-3, IL-7, IL-15, and Flt3L, which favors the development of NK cells. Another important factor for the differentiation of NK cells from ESCs or iPSCs in vitro is the HOXB4 homeoprotein. Larbi et al. [46] demonstrated that HOXB4 can promote the expansion of EB-derived hematopoietic precursors that can differentiate into mature and functional NK cells. For clinical applications, the next step is the production of clinical-scale NK cells derived from ESCs or iPSCs. Knorr et al. produced iPSCs-derived NK cells with an improved two-stage method that does not involve xenogeneic stromal cells and for which cell sorting of the NK cells is not required.

While there are some critical advantages of CAR-T cell therapy, immunotherapy with CAR-NK cells can offer some alternative approaches to overcoming the challenges associated with CAR-T cells [47]. Several studies demonstrated that CAR-NK cells may revolutionize gastrointestinal cancer therapy, offering a pathway toward future clinical implementation [48]. Significantly, CAR-NK cells exhibit key benefits, including minimized graft-versus-host disease (GVHD) risk, attenuated cytokine release syndrome (CRS), and dual targeting of cancer cells through CAR-mediated and natural cytotoxicity mechanisms.

Additionally, NK-92, an NK cell line, is gaining the interest of researchers as an alternative to primary human NK cells. In 2008, Arai et al. [49] initiated a Phase I trial in which the allogeneic cell line NK-92 was infused into patients with advanced renal cell cancer or melanoma. This study demonstrated that it is feasible and safe to administer NK-92 cells for allogeneic cellular immunotherapy of advanced renal cell cancer or melanoma. Recently, NK-92 cells were used for refractory hematological malignancies relapsing after autologous hematopoietic cell transplantation (AHCT) in a Phase I trial [50]. In this study, the final NK-92 cells were resuspended in GM2 medium and irradiated with 10 Gy. The results showed that, in this trial, two patients had a complete response, two patients had minor responses, and one had clinical improvement. This study demonstrated that it is feasible and safe to administer irradiated NK-92 cells at very high doses in patients with refractory hematological malignancies relapsing after AHCT. Preclinical NK-92 cells have been engineered to express antigen receptors and surface protein receptors to CD19, HER2, and EGFR [51]. Although it has not been reported in clinical trials with CAR NK-92 cells, preclinical studies demonstrate that researchers engineered CAR-NK-92 cells with CRISPR-Cas9 to target acute myeloid leukemia (AML) and B cell acute lymphocytic leukemia (B-ALL) [52].

It has been demonstrated that human iPSCs-derived NK cells have abilities similar to those of NK cells derived from peripheral blood against ovarian cancer [53]. In this study, iPSCs derived from CD34+ umbilical cord blood cells were induced into NK cells via the formation of embryoid bodies. Then, the researchers investigated antitumor activity in a mouse xenograft model. The experimental results showed that the treatment with the aAPC-expanded iPSCs-derived NK cells was better than that of the aAPC-expanded NK cells derived from peripheral blood. Recently, Ghobadi et al. demonstrated that gene-modified, iPSCs-derived NK cells represent a powerful cell therapy platform for cancer, potentially overcoming key limitations of current immune cell therapies such as manufacturing time, product heterogeneity, accessibility, and cost [54]. More clinical studies are needed for the future application of iPSCs-derived NK cells in the clinic.

### 3.2. iPSCs-Macrophages

Macrophages constitute a population of bone marrow-derived myeloid cells that circulate in the blood as immature monocytes [55]. Macrophage populations that inhibit the growth of cancer cells or induce cancer cell death are now called M1 macrophages, and those that promote the growth or repair of cancer cells are called M2 macrophages [56]. Recently, Beatty et al. [57] demonstrated that the CD40 pathway can be activated to harness the cancer immune surveillance of macrophages in both mice and men. In their study, this group demonstrated that CD40-activated macrophages can rapidly infiltrate tumors, become tumoricidal, and facilitate the depletion of tumor stroma in a mouse model of pancreatic tumors.

Currently, experimental macrophages are derived mainly from two sources: tumor-derived cell lines (e.g., U937 and THP-1 cells) and primary cells, such as microglia, alveolar macrophages, and human monocyte-derived macrophages (HMDMs) [58]. THP-1 cells have an unlimited potential to replicate, but their karyotypes are abnormal, and their functions are immature. Primary macrophages, especially those from healthy subjects, have limited self-renewal ability and availability. iPSCs are genotype-specific, scalable, and self-renewable sources for use in generating macrophages for cell-based cancer immunotherapy. iPSCs-derived macrophages can be generated via a protocol based on the OP9 coculture system or the formation of embryoid bodies [59]. Both of these protocols require the addition of GM-CSF or M-CSF to induce their differentiation into macrophages. Recently, Monkley et al. [60] developed a serum-free culture system to culture iPSCs and induce monocyte-like cells to differentiate into macrophages via medium with serum and M-CSF. Although iPSCs-derived macrophages have phenotypic, functional, and transcriptomic characteristics similar to those of HMDMs, their ontogeny and identity have not been defined, and further studies are needed for their application in future cell-based cancer immunotherapy.

### 3.3. iPSCs-T Cells

Currently, there are various methods to produce T cells used in cancer immunotherapy. In the 1980s, the group of Steven Rosenberg initially used IL-2 to activate lymphocytes in vitro and endowed these lymphocytes with cytotoxic properties [61]. Autologous lymphokine-activated killer (LAK) cells, together with or without systemic IL-2, were used in cancer immunotherapy [62]. To selectively optimize the activated immunocytes, some studies utilized tumor-infiltrating lymphocytes (TILs) for cancer immunotherapy. In animal studies, it was shown that TILs have 50- to 100-fold more effective therapeutic activity than shown by LAK cells [63]. Recently, due to the promising results of autologous expanded TILs, attempts were made to redirect T cells by transfecting them with TCR, which recognizes a specific tumor antigen or chimeric antigen receptor (CAR) [64].

It has been proven that hematopoietic stem cells (HSCs) and embryonic stem cells (ESCs) can differentiate into lymphocyte lineage cells in the OP-9 coculture system in vitro [65]. Similarly, Lei et al. [66] induced iPSCs-produced mouse embryonic fibroblasts to differentiate into a lineage in an OP9-DL1 coculture system. In an important research advancement, iPSCs can be successfully generated from primary CD34+ hematopoietic progenitor cells obtained from peripheral blood [67]. However, due to the low number of hematopoietic progenitor cells in peripheral blood, various studies have generated iPSCs with PBMCs [68,69]. To apply iPSCs-derived T cells to cancer immunotherapy, several studies have exploited multiple methods for generating T cells from human iPSCs. Iriguchi et al. demonstrated enhanced T cell differentiation efficiency using either antigen-specific cytotoxic T cell clone-derived iPSCs or TCR-transduced iPSCs as starting materials [70]. Some studies applied iPSCs derived from CTLs containing a particular epitope for the generation of antigen-specific T cells [71]. In recent years, to improve the tumor targetability of iPSCs-derived T cells, the combination of iPSCs generation technology with the transduction of tumor antigen-specific T cell receptors (TCRs) or chimeric antigen receptors (CARs) has become an effective strategy for generating tumor-specific T cells [71].

Of course, there are some deficits in using iPSCs-derived T cells. First, iPSCs-derived T cells confer a risk of teratoma genesis, although it has been found in only one Rag−/− mouse [40]. Second, the immunogenicity of iPSCs-derived T cells is very complicated, and immune rejection is T cell-dependent [72,73]. Finally, it has been widely accepted that the reprogramming process used for generating iPSCs can induce both genetic and epigenetic defects [74,75].

Another promising method of immunotherapy involves the use of chimeric antigen receptor T (CAR-T) cells, which are made by transfecting T cells with TCR that can recognize tumor-specific antigens [63]. One of the first clinical trials of CAR-T cells was reported by the June group in 2011 [76]. In this study, CAR-T cells were designed to target the B cell- and chronic lymphocytic leukemia (CLL)-specific protein CD19. The therapy was considered effective as two of three patients experienced complete remission, confirming that CD19 in both bone marrow and blood was responsive.

In the case of solid tumors, Shabaneh et al. [77] demonstrate that high-affinity CARs targeting antigens like HER2 (which are expressed on normal tissues) pose significant safety risks. Conversely, low-affinity HER2 CARs can effectively induce tumor regression while maintaining safety, suggesting a viable therapeutic approach for HER2-overexpressing solid tumors. Recently, Katz et al. [78] initiated a study in which patients with CEA+ liver metastasis were injected with CAR-T cells through percutaneous hepatic artery infusion (HAI). The patients enrolled in this study received systemic chemotherapy before treatment with the anti-CEA CAR-T cells. The results showed that the patients who received systemic IL-2 along with anti-CEA CAR-T cells had a more favorable CEA response to the treatment. This study demonstrated the safety of anti-CEA CAR-T HAI in a heavily pretreated population with an extensive tumor burden.

As described above, Themeli et al. [79] demonstrated that human iPSCs can be combined with CAR technologies to generate human T cells targeted to CD19. In this study, iPSCs were reprogrammed from peripheral blood lymphocytes via retroviral vectors and transduced with a lentiviral vector encoding 1928z, a second-generation CAR. Then, these 1928z-T-iPSCs were induced to redifferentiate into T cells via the formation of embryoid bodies in an OP9 coculture system. The experimental results showed that iPSCs-derived CAR-expressing T cells displayed a phenotype similar to that of innate γδ T cells and induced high antigen-specific cytotoxicity in vitro. Finally, the researchers investigated the antitumor activity of 1928z-T-iPSCs-T cells in a xenogeneic tumor model. The experimental results showed that iPSCs-derived T cells potentially inhibited tumor growth in vivo, resembling that of CAR-transduced peripheral blood γδ T cells. Furthermore, some groups have already set up clinical trials for the future clinical application of iPSCs-derived immunotherapy. Kawamoto’s group is working to set up a clinical trial using iPSCs-derived T cells that target the WT 1 antigen in acute myeloid leukemia (AML) patients [8]. Aged AML patients who have not received stem cell transplants and experienced relapse after chemotherapy, and for whom no effective treatment is available, will be admitted into this clinical trial.

### 3.4. iPSCs-Dendritic Cells (DCs)

Conventional cell types for the production of DC vaccines are composed of monocytes [80], CD34+ hematopoietic progenitor cells derived from bone marrow [81], peripheral blood [82], or cord blood [83]. Among these conventional cell types, monocytes are the most commonly used in the production of DC vaccines. In 2000, Fairchild et al. [84] first reported the application of embryonic stem cells for the generation of dendritic cells. In contrast to DCs generated by monocyte-based methods, DCs derived from embryonic stem cells can activate a more powerful immune response [85]. However, any method using embryonic stem cells may not be genetically compatible for every patient, and ethical issues limit the application of embryonic stem cells to the generation of DC. These two problems can be solved by using induced pluripotent stem cells (iPSCs) [86].

iPSCs derived from both human and mouse somatic cells can be used for the generation of DCs. DCs derived from iPSCs present antigens, stimulate the proliferation of T cells, and produce cytokines, and DNA analyses show that the DCs derived from iPSCs have similar upregulated levels of genes relative to antigen presentation to the DCs derived from bone marrow [87]. Currently, the method for the generation of iPSCs-derived DCs in vitro is categorized into two types: one is based on embryoid body-mediated hematopoietic differentiation, and the other is based on OP9 coculture-based hematopoietic differentiation. Both methods require granulocyte/macrophage colony-stimulating factor (GM-CSF), which plays a vital role in the generation of iPSCs-derived DCs. However, iPSCs-derived DCs generated in culture medium containing fetal calf serum and the OP9 coculture system cannot be used in the clinic. Therefore, in recent years, researchers have developed xeno-free culture systems to maintain iPSCs and generate iPSCs-derived DCs [88,89].

iPSCs-derived DCs generated with xeno-free culture systems have similar phenotypes to DCs derived from monocytes (monocyte-derived DCs, moDCs), which can induce the proliferation of T cells with which they are cocultured and secrete high levels of proinflammatory cytokines under inflammatory conditions [89]. In contrast to moDCs, iPSCs-derived DCs generated with a xeno-free culture system have the CD11bloCD141hi phenotype and a subtype that stably expresses CD141+ XCR+. This subtype has highly specific cross-antigen-presenting functions [89]. However, the xeno-free culture system is still not perfect and cannot be applied to all types of iPSCs clones in the same way as conventional cultivation systems.

The most advanced DC-based immunotherapy is Provenge, which was approved by the FDA for the treatment of androgen-resistant prostate cancer [90]. Provenge is based on monocyte-derived DCs, which are cultured with a chimeric protein composed of GM-CSF and prostate-specific antigen (PSA). Recently, a Phase I study was initiated in which WT 1-pulsed dendritic cell vaccines were given to treat unresectable advanced pancreatic ductal adenocarcinoma [91]. In this study, dendritic cells were pulsed with the WT 1 peptide to improve the targeting of the CTLs. Patients were treated with 1 cycle of nab-paclitaxel/gemcitabine chemotherapy alone, followed by 15 administrations of the WT1-DC vaccine without concurrent chemotherapy. The chemoimmunotherapy regimen, including WT1-DC vaccine, potently activates WT1-specific immunity in unresectable advanced pancreatic ductal adenocarcinoma patients, potentially remodeling the tumor microenvironment to facilitate conversion surgery and improve clinical outcomes.

It was recently demonstrated that intratumoral administration of dendritic cells derived from induced pluripotent stem cells (iPSCs-DCs), combined with local radiotherapy, synergistically enhances systemic antitumor immunity in poorly immunogenic mouse tumor models resistant to PD-L1 blockade [92]. In this study, the iPSCs-DCs phenotypically resemble conventional type 2 DCs and effectively prime antigen-specific CD8+ T cells in vitro. In vivo, the combination of iPSCs-DC injection and RT improved tumor-specific CD8+ T cell priming, increased trafficking of injected iPSCs-DCs to tumor-draining lymph nodes, promoted the formation of DC/CD8+ T cell aggregates, and enriched the tumor microenvironment (TME) with stem-like Slamf6+ TIM3− CD8+ T cells. Notably, this combinatorial treatment delayed the growth of both treated tumors and untreated distant tumors, indicating the induction of systemic antitumor immunity. Furthermore, the combined therapy upregulated PD-L1 expression on tumor-associated macrophages and DCs, thereby sensitizing previously resistant tumors to anti-PD-L1 immune checkpoint blockade. Addition of anti-PD-L1 therapy to the regimen significantly improved tumor control and survival and led to the development of tumor-specific immunological memory, as mice that achieved durable tumor regression rejected subsequent tumor rechallenge. These findings highlight the translational potential of using iPSCs-DCs in combination with RT to overcome resistance to immune checkpoint therapy in poorly immunogenic cancers by creating a more favorable immunological milieu that enhances T cell priming, infiltration, and function. Moreover, it will be indispensable for clinical trials on human iPSCs-derived DCs used for cancer immunotherapy in the future.

### 3.5. iPSCs-B Cells

It is well known that different subtypes of B lymphocytes have distinct immunological functions. For example, IgA (+) B cells play a role in mucosal immunity, while IgG (+) B cells are involved in humoral immunity [93]. However, the mechanism of B cell functions in the antitumor response is not clear and has led to contradictory findings [94]. Some experiments have suggested that B cells can enhance antitumor immunity, while others have indicated that B cells disrupt the antitumor response. Thus, B cells play a dual role in antitumor immunology [95]. Recently, Yiwen et al. [96] expanded human B lymphocytes ex vivo and potently enhanced their antigen-presenting ability via a coculture system containing both CD40 L and B-cell activating factor (BAFF). Their findings indicated that antigen-specific CTLs can be generated using this type of B cell as an APC, and this approach can be applied to CTL-mediated immunotherapy in cancer patients.

Currently, iPSCs have not been widely used in the generation of B cells for cancer immunotherapy. Jiang et al. [97] indicated that iPSCs-derived B cells can be used in antitumor immunotherapy in a cell-based vaccine. Some other studies have demonstrated that B cells can be generated from iPSCs; however, these iPSCs-derived B cells have not been used in antitumor immunotherapy [98].

## 4. Improving the Differentiation of Immunocytes: Small Molecules, Nanoparticles and 3D Culture Systems

The conventional methods for modulating the differentiation of iPSCs in vitro involve the application of growth factors and cytokines. However, there are various drawbacks associated with these conventional methods, such as the optimization of the culture and medium that does not induce toxicity, high costs, and unidentified mechanisms of action [99]. In contrast, small molecules can work within the cell by modulating a specific protein or targeting a specific protein family. The effects of small molecules are usually reversible, which allows small molecules to modulate protein functions and signals more flexibly. Compared with conventional methods, small molecules are inexpensive to purchase and easier to store [100]. Zhu et al. [101] applied a small molecule-based protocol using IWR1, SB431542, LDN193189, and IGF1 to induce the differentiation of iPSCs into transportable retinal photoreceptors. iPSCs can also be induced into hepatocytes via small molecules, which can deplete hepatocyte nuclear factor 4 alpha (HNF4A) [102]. Although there has been no report on iPSCs-derived immunocytes generated through a small molecule protocol, this strategy must be applied to the differentiation of iPSCs-derived immunocytes in the future.

Nanoparticles are also promising tools for improving the differentiation of iPSCs. Based on their small size and large surface area, nanoparticles can readily enter the cell and mediate various changes in molecules [103]. An ordered nano-topographical groove combined with peptides can promote the differentiation of iPSCs into osteogenic cells [104]. iPSCs grown on carbon nanotubes (CNTs) can differentiate into mesoderm; in contrast, iPSCs grown on conventional culture dishes tend to differentiate into ectoderm [105].

Stem cells may have different characteristics in the human body compared with those manifested in vitro. The main reason for the difference is that most of the studies on stem cells are carried out in 2D cell culture systems, such as multi-well plates, Petri dishes, and coverslips [106]. Cells cultured in these 2D environments usually display an irregular morphology and abnormal cell-to-cell interactions [107]. Thus, it is very important to establish a cell-based culture system in vitro that can simulate the physicochemical environment of the extracellular matrix (ECM) in vivo. In contrast to a 2D culture system, a 3D culture system not only can mimic the environment of the ECM but also reflect the actual state of cells in a 3D microenvironment, including differences in their morphology, migration, proliferation, and differentiation [108].

In the 1980s, dedifferentiated chondrocytes were encapsulated in a 3D culture system, and researchers found that this 3D culture system restored the phenotype of chondrocytes, including their shape and the expression of cartilaginous markers [109]. Subsequent observations revealed that the dimensions of a culture system are crucial factors influencing the morphology and function of cells. Cells cultured in a monolayer culture system were inclined to have abnormal dedifferentiation functions, whereas those cultured in a 3D system were inclined to have a more typical physiological state [110]. In a 3D culture system, cells are usually grown in a cell-to-cell or cell-to-scaffold environment, which is closer to the in vivo environment than that of a 2D culture system and more suitable for cell growth, migration, proliferation, and differentiation. Over the last three decades, enormous efforts have been made to develop various 3D culture systems, and two main types have emerged [111]. The first type is based on scaffold-free 3D cell spheroids, which can be generated from suspensions of cells subjected to external physical forces. The second type is a scaffold-based 3D culture system, which is based on seeding cells on an acellular porous 3D scaffold. This type of 3D culture system mimics the structure of the ECM and the in vivo environment.

Recently, 3D culture systems have been used in the field of iPSCs. It has been reported that iPSCs cultured in a 3D scaffold composed of alginate, carboxymethyl chitosan, or agarose can differentiate into embryoids composed of endoderm, ectoderm, and mesoderm or can differentiate into more homogeneous neural tissues [112]. Seet et al. [113] showed that mature T cells can be generated from human hematopoietic stem/progenitor cells via artificial thymic organoids. Nevertheless, most of the studies are limited to in vitro research or the sub-organ level, and there is more work to be conducted for the application of 3D stem cell cultures in the clinic.

## 5. Improving the Expansion of Immunocytes In Vivo

iPSCs-derived immunocytes need to be expanded for application in the clinic. The immunocytes derived from patients can be expanded in vitro because this approach is simpler and more established (Figure 2). The expansion of T cells in vitro can be achieved by mimicking normal T cell activation [114]. In vivo, autonomous T cells can be activated to produce IL-2, which is pivotal for the expansion of clones. Similarly, T cells can be expanded with a boost of IL-2. Thus, IL-2 is called T cell growth factor. However, there are some drawbacks to the method based on IL-2 in vitro. T cells induced by IL-2 are inclined to differentiate into effectors at the expense of the formation of memory T cells [115]. Boyman showed that IL-2 can affect the development of CD8 memory T cells [116]. Additionally, excessive stimulation during in vitro expansion may contribute to the exhaustion of T cells, which leads to fewer T cells being effective during therapy. Several approaches have been proposed to improve the expansion of T cells. Chen et al. [117] demonstrated that IL-7 can selectively sustain the expansion of CTLs without enhancing Tregs in vitro or in vivo.

Granulocyte-macrophage colony-stimulating factor (GM-CSF) was first reported to sustain the expansion of DCs in vitro [118]. Another important factor modulating the expansion of DCs is FMS-like tyrosine kinase 3 ligand (Flt3L). The unique role of Flt3L, compared with that of GM-CSF, in the differentiation of DCs was first recognized in human bone marrow culture in vitro [119] and was later confirmed in mouse bone marrow culture [120]. The important difference between DCs developed in Flt3L culture and those developed in GM-CSF culture is that the former are mostly resident DC subsets, and the latter are inflammatory DCs [121]. Flt3L can also enhance the expansion of DCs in vitro [122], while the DCs generated from Flt3L culture lack the full expression capacity of the resident spleen DCs, as indicated by marker proteins (e.g., deficiency of CD8α) [121]. Lam et al. [123] engineered a novel FLT3L-FC DNA construct that can induce robust expansion of DCs in vivo. iPSCs-derived immunocytes can also be expanded with in vivo approaches for future applications in the clinic, although no such efforts have been reported.

## 6. Overcoming the Obstacle of Cancer Targeting Disability in iPSCs-Derived Immunocytes: The Optimization of Antigen Specification and Presentation

### 6.1. APCs

The cycle of immune responses to tumors is initiated by antigen-presenting cells (APCs), such as DCs and B cells, to capture tumor-associated antigens. Many methods have been explored to induce APC capturing of TAAs. The classical method involves adding DCs cultured ex vivo into tumor cell lysates, obtaining total tumor RNA, or generating synthetic antigenic peptides [124]. Mu et al. [125] used the tumor cell lysates of patients to pulse autologous DCs and then transfer autologous vaccine-primed T cells in combination with the vaccine. The experimental results showed that this method is safe and feasible, and patients with recurrent ovarian cancer can benefit from this method. This strategy can also be used in iPSCs-derived dendritic cells. Zhang et al. [126] demonstrated that iPSCs-derived DCs pulsed with H-2Kb-restricted OVA peptide in vitro can prime OVA-specific CTLs and elicit protective action in mice with OVA-expressing melanoma.

Compared with the classical method of inducing DCs to capture TAAs, gene-editing technology, such as that based on vectors or synthetic mRNA, offers a more robust means to introduce antigens into DCs; specifically, antigens can be translated and presented by the endogenous antigen-processing DC machinery. It has also been shown that DCs can be engineered to express MART-1, a type of tumor-associated antigen, via an adenoviral vector, which DCs can present to T cells to induce specific CTLs against MART-1 [127]. Gene-editing technology can be applied not only to primary DCs but also to DCs derived from iPSCs. Furukawa et al. [128] demonstrated that iPSCs-derived DCs genetically modified via recombinant adenoviral vectors can induce the generation of CEA-specific CTLs. In addition to its application to tumor-associated antigens, gene-editing technology can also be used with tumor neoantigens. RNA-sequencing technology can be used to identify the expression of tumor neoantigens in tumors. The application of gene-editing technology enables the expression of these neoantigens on DCs, and based on these neoantigens, the DCs induce the production of CTLs against tumors [129]. In addition, the application of gene-editing technology can also enhance the expression of positive regulators on DCs. The CD40 ligand (CD40L) is a potent activator of DCs. Electroporation of mRNA [130] or transduction with adenovirus [131], lentivirus [132], or vaccinia virus vectors [133] can be used to engineer DCs to express CD40L, which can enhance the expression of costimulatory molecules (CD80 and CD86) on DCs. All of these factors can enhance the ability of DCs to present tumor antigens.

### 6.2. CARs

The chimeric antigen receptor (CAR) or tumor-specific cell receptor produced by gene editing can be expressed in immune effector cells [114], which can improve the targeting of effector immunocytes. Researchers have used a variety of vectors for the transfection of CAR, including lentiviral vectors and retrovirus vectors, and the Sleeping Beauty transposon system [114]. Currently, highly efficient gene transfection systems based on murine retrovirus or lentiviral vectors have become the most commonly used methods of gene transfection for use in mammalian gene therapy [134]. In addition to its application in primary immunocytes, CAR engineering can also be used in immunocytes derived from iPSCs. Themeli et al. [79] demonstrated that CAR engineering can be combined with iPSCs-derived immunocytes via retroviral and lentiviral transduction. In this study, peripheral blood lymphocytes were first reprogrammed into T-iPSCs by retroviral transduction. Then, these T-iPSCs were transduced with a lentiviral vector encoding 19-28z, a second-generation CAR. Finally, these 1928z-T-iPSCs were used to induce T cells targeting CD-19 for cancer immunotherapy. The experimental results showed that these iPSCs-derived, CAR-expressing T cells had a phenotype resembling that of innate γδ T cells and potently inhibited the growth of tumors in a xenograft model.

There are some drawbacks to retroviral and lentiviral transduction. Retroviruses can efficiently transfect only dividing cells and can easily be integrated into promoters of the host genome, which leads to the aberrant expression of genes and oncogenicity [135]. The risk of inserting mutations is lower with lentiviruses than it is with retroviral vectors [136]; however, the current data indicate that T cells genetically modified by lentiviruses cannot persist for a long time because of the dysregulation of their proliferative and survival pathways [137]. To improve the efficacy of transgenic integration into modified T cells, researchers have begun to investigate the transposons and transferases of the Sleeping Beauty (SB) system [138]. In this system, two DNA plasmids are transfected into T cells via electroporation. One of the plasmids carries the desired transgene, and the other carries the transferase. Monjezi et al. [139] demonstrated that CAR T cells can be engineered through non-viral SB transposition of CAR genes from minimalistic DNA vectors. In addition, the results showed that, compared with the effect of lentiviral vectors, the electroporation used in the SB system is less mutagenic and genotoxic. CD19-expressing CAR T cells engineered through electroporation using the SB system conferred potent reactivity in vitro and eradicated lymphoma in the xenograft model in vivo. It has been demonstrated that the SB transposon system can also be used in the reprogramming of iPSCs, and this strategy allows for the targeted gene insertion in patient-derived iPSCs [140]. Thus, the SB transposon system has promise for use in iPSCs-derived CAR T cells in the future.

Recently, three classes of DNA-editing nucleases, zinc-finger nucleases (ZFNs), transcription activator-like effector nucleases (TALENs), and RNA-guided clustered regularly interspaced short palindromic repeats (CRISPR) systems have been applied to cancer immunotherapy. TALENs and ZFNs are based on the combination of the nuclease domain of the Fokl restriction enzyme and the respective TALE or ZFN DNA-binding domain. Engineered TALENs and ZFNs can be used to recognize a wide range of DNA sequences. However, two TALEN and ZFN monomers are required to initiate the cleavage of DNA because the nuclease domain of the Fokl restriction enzyme functions as a dimer, which potentially increases both the specific and off-target effects due to the tendency for dimer mismatches [141]. The CRISPR system has the following advantages over the two gene-editing techniques. First, the CRISPR system requires only a short complementary sgRNA for DNA targeting, and the synthesis of the complementary sgRNA is relatively simple and cost-effective. Obviously, through the application of a variety of sgRNAs targeted for different sequences, the CRISPR system can edit multiple independent targeted sites simultaneously, which provides a high-throughput method for gene editing [142]. Additionally, there are several variants of the Cas9 protein, which can be used for editing multiple genes; for example, the epigenome can be targeted by catalytically deactivated Cas9 (dCas9) [143]. Of course, the CRISPR system has limitations and needs to be improved. The primary challenges to using CRISPR are the potential off-target effects. Several methods have been applied to decrease the off-target effects of the CRISPR system, such as the use of the Cas9 nickase mutant [144], the catalytically inactive Cas9 fused to the FokInuclease [145], a reduced concentration of Cas9 protein in the CRISPR system [146], the newly the developed Cas9 variants eSpCas9 and SpCas9-HF1 [147], and the CRISPR/Cpf1 system [148].

The CRISPR system can improve the targeting of therapeutic immune cells because of its simple design and flexibility; therefore, the CRISPR system can be widely used in cancer cell-based immunotherapy. Among the applications of the CRISPR system to cancer cell-based immunotherapy, the most attractive is the production of chimeric antibody receptor (CAR) T cells via the regulation of gene expression to identify specific antigens on cancer cells. Anti-CD19 CAR-T cells engineered via the CRISPR system have exhibited a unique effect against nonleukemic disease and leukemia [149]. Novartis applied an approved Intellia CRISPR-cas9 gene editing system used to design CAR-T cells for treatment [150]. Additionally, the CRISPR system can be used in a wide range of iPSCs-derived cell types. Mandegar et al. [151] demonstrated that clustered regularly interspaced short palindromic repeat interference (CRISPRi) can specifically and reversibly inhibit gene expression in iPSCs and iPSCs-derived cardiac progenitors, cardiomyocytes, and T lymphocytes. A CRISPR system can be feasibly used to design iPSCs-derived CAR-T cells for future cancer immunotherapy.

In addition to combining CAR engineering with T cells, CAR engineering can also be used with hematopoietic cells, cytokine-induced killer cells (CIK), monocytes, neutrophils, and NK cells [51]. Because of the inherent lytic potential of NK cells, expressing CARs in NK cells is appealing [152]. Laskowski et al. [153] demonstrated that the expression of CS1-CAR redirected NK cells to eradicate CS1-expressing multiple myeloma cells both in vitro and in vivo. In addition to its effect on primary cells, CAR engineering can be used with iPSCs-derived immunocytes. Themeli et al. [79] demonstrated that CAR engineering can be combined with iPSCs-derived T cells targeted to CD-19 for cancer immunotherapy. With the continuous optimization of the protocol for inducing iPSCs into NK cells, CAR engineering will also be used in iPSCs-derived NK cells in the future.

## 7. Enhancing the Killing Effectivity: Modulating the Tumor Immune Microenvironment and Anti-Apoptosis Mechanisms

### 7.1. Persistence

One approach to improving the persistence of immunocytes is to alter the host environment into which the immune cells are infused. For example, the use of chemotherapy or radiotherapy to reduce or deplete lymphocytes in recipients can improve the persistence of adoptive immunocytes [154]. Simultaneously, it has been demonstrated in animal studies that this method can also enhance the persistence and efficacy of CAR-T cells in vivo. However, combining chemotherapy with cell-based immunotherapy may increase the toxicity of the treatment and compromise the antitumor effect of the cell-based immunotherapy. Another approach is to deliver an activation signal with the intracellular domain of CAR to maintain the proliferation of the immunocytes. For example, incorporating CAR with CD28 or 4-1 BB stimulatory domains can induce immune cells to secrete more IL-2, increase the proliferation of T cells, and mediate a greater antitumor response [155]. In addition to CD28 and 4-1 BB, many other costimulatory domains can also be incorporated into CARs to further enhance the effects of their co-stimulation, including CD27, OX40, ICOS, CD40, and MyD88 [156]. The third approach is to prolong the replicative lifespan of human immunocytes. The application of retrovirus vectors can prolong the length of telomeres in T cells, which is an attractive strategy for use in adoptive immunotherapy [157]. The fourth method is to regulate the exhaustion of T cells. It has been demonstrated that exhausted human and mouse T cells have sustained expression of PD-1 [158]. Genetic depletion of the PD-1 receptor on T cells can effectively enhance T cell-based cancer immunotherapy. Chow et al. [159] applied electroporation technology to transfer sgRNA and Cas9-encoding plasmids into human T cells, resulting in a significant reduction in PD-1 expression via the CRISPR gene-editing system, and they found that this strategy can enhance the immune efficacy of T cells and their cytotoxic effect on cancer cell lines. iPSCs are novel cell sources for cell-based cancer immunotherapy, and the above approaches can also be applied to iPSCs-derived immunocytes.

### 7.2. Trafficking

To function within a tumor, genetically modified immunocytes must have the ability to home to the malignancy. In the process of gene editing and passaging in vitro, migration can be compromised due to the loss of desired cytokine receptors, or immunocytes that cannot inherently move to certain tissues are selected. It has been shown that using lentiviral vectors to introduce the CCR2 receptor into CAR-T cells can improve their ability to migrate into the tumor and enhance the antitumor effect of the CAR-T cells in vivo [160]. However, overall, relatively few works have focused on improving the trafficking of immunocytes. Perhaps gaining a greater understanding of how to improve the chemotaxis and migration of immunocytes requires a novel approach.

### 7.3. Overcoming the Mechanisms of Resistance

Perhaps the most needed advance for the use of engineered antitumor immunocytes is the ability to overcome or remodel the immunosuppressive microenvironment, particularly in solid tumors. The first approach involves engineering immunocytes to ignore the immunosuppressive signal. For example, the application of a retrovirus vector to downregulate the expression of the TGF-β receptor on T cells can reduce the inhibitory effect of TGF-β on T cells [161]. Similarly, PD-1 is a checkpoint inhibitor expressed on the surface of tumor cells that can bind to the PD-1 receptor on activated T cells and inhibit the killing effect of cancer mediated by T cells. Lowering the expression of PD-L1 on immunocytes can also reduce the inhibitory effect of the immune microenvironment on immunocytes. By using lentiviral vectors, the extracellular PD-1 domain is fused with the intracellular costimulatory domain to direct the PD-L1 signal that inhibits T cells under normal conditions to activate T cells [162]. As described above, using CRISPR technology to reduce the expression of PD-1 on T cells can also reduce the inhibitory effect of PD-L1 in the immune microenvironment. This approach is also applicable to iPSCs-derived immunocytes. NK cells express the low-affinity Fc-activating receptor CD16 and the inhibitory receptor killer cell immunoglobulin-like receptor (KIR). Qin et al. [163] attempted to establish an iPSCs-derived NK cell line by knocking out or knocking down the KIR gene to reduce the immunosuppressive effect of the immune microenvironment on iPSCs-derived NK cells. Another general strategy involves enabling immunocytes to remodel the immune microenvironment. IL-12 is among the most promising antitumor cytokines and can play a variety of roles in both innate and adaptive immunity; thus, it is a very effective agent to use for remodeling the immune microenvironment. For example, the application of retroviral vectors can make CAR-T cells constitutively express the potent IL-12 cytokine [164]. Recently, using a lentivirus vector, the synNotch receptor system was introduced into engineered T cells to induce T cells to secrete a specific payload that recognizes the targeted antigens [165]. The synNotch receptor system can carry a variety of desired payloads, including IL-12, proinflammatory cytokines, checkpoint antibodies, and bispecific antibodies. Therefore, in principle, immunocytes can be programmed to be delivery agents or “pharmacytes”.

## 8. Future Directions and Conclusions

### 8.1. iPSCs in Cell-Based Cancer Immunotherapy

Induced pluripotent stem cells (iPSCs) have emerged as a transformative tool in the field of cell-based cancer immunotherapy, offering a versatile platform for generating various types of immunocytes. These include dendritic cells (DCs) for cancer vaccines, natural killer (NK) cells, chimeric antigen receptor T (CAR-T) cells, and B cells. Each of these cell types plays a crucial role in the immune response against cancer, making iPSCs a valuable resource for developing novel therapeutic strategies.

Despite the significant potential of iPSCs, several challenges remain. One of the primary concerns is the risk of tumorigenicity, where iPSCs may form tumors instead of differentiating into the desired cell types. Additionally, the efficiency of reprogramming somatic cells into iPSCs is often low, and the differentiation processes required to generate specific immunocytes are costly and complex. These limitations have spurred the development of new strategies to enhance the efficiency and safety of iPSCs-based therapies.

### 8.2. Overcoming Limitations with Innovative Techniques

Recent advancements in small-molecule and nanoparticle technologies have shown promise in addressing some of these challenges. Small molecules can modulate specific cellular pathways to enhance the efficiency of iPSCs reprogramming and reduce the risk of tumorigenicity. For example, certain small molecules can activate key transcription factors involved in pluripotency, thereby improving the yield and quality of iPSCs. Nanoparticles, on the other hand, can deliver essential factors directly to cells, facilitating more efficient differentiation into the desired immunocyte types.

Three-dimensional (3D) culture systems have also proven to be a significant advancement in iPSCs differentiation. Unlike traditional 2D cultures, 3D systems provide a more physiologically relevant environment that mimics the in vivo conditions. This leads to better differentiation outcomes, resulting in immunocytes with enhanced functionality. The use of 3D cultures not only improves the quality of the differentiated cells but also reduces the overall costs associated with large-scale production, making iPSCs-derived immunocytes more accessible for therapeutic applications.

In vivo techniques have further expanded the potential of iPSCs by increasing their expansion efficiency. These methods allow for the generation of larger quantities of immunocytes, which is crucial for clinical applications where high cell numbers are required. Additionally, in vivo expansion can occur in a more controlled and physiologically relevant environment, further enhancing the functionality and safety of the derived cells.

The CRISPR system has revolutionized the precision with which iPSCs-derived immunocytes can be targeted to cancer cells. By enabling precise genetic modifications, CRISPR technology enhances the specificity and efficacy of immunotherapy, reducing off-target effects and improving patient outcomes. This has laid a solid foundation for the future clinical application of iPSCs-derived immunocytes, making them a more viable option for cancer treatment.

### 8.3. Clinical Applications and Future Prospects

The integration of iPSCs into cancer immunotherapy holds the potential to transform the treatment landscape for cancer patients. iPSCs-derived immunocytes offer several advantages over traditional cell therapies, including their ability to be produced in large quantities, their potential for off-the-shelf availability, and their reduced risk of immune rejection. These features make iPSCs-based therapies particularly attractive for widespread clinical use.

However, despite these advancements, the field is still in its early stages. Further research is needed to optimize reprogramming and differentiation protocols, enhance the safety and efficacy of iPSCs-derived immunocytes, and address the ethical and regulatory challenges associated with their use in clinical settings. Clinical trials are essential for validating the therapeutic potential of iPSCs-based immunotherapies and for ensuring their safe and effective application in cancer treatment.

### 8.4. Conclusions

In summary, iPSCs-based cancer immunotherapy represents a groundbreaking approach with the potential to revolutionize cancer treatment. While significant progress has been made in overcoming technical challenges, continued research and development are necessary to fully realize the clinical potential of iPSCs-derived immunocytes. As we move forward, the collaboration between researchers, clinicians, and regulatory bodies will be crucial in translating these innovative therapies from the laboratory to the clinic, ultimately providing new hope for cancer patients. The future of iPSCs-based immunotherapy looks promising, but it will require sustained effort and innovation to overcome the remaining hurdles and bring these therapies to the forefront of cancer treatment.

## Figures and Tables

**Figure 1 biomedicines-13-02012-f001:**
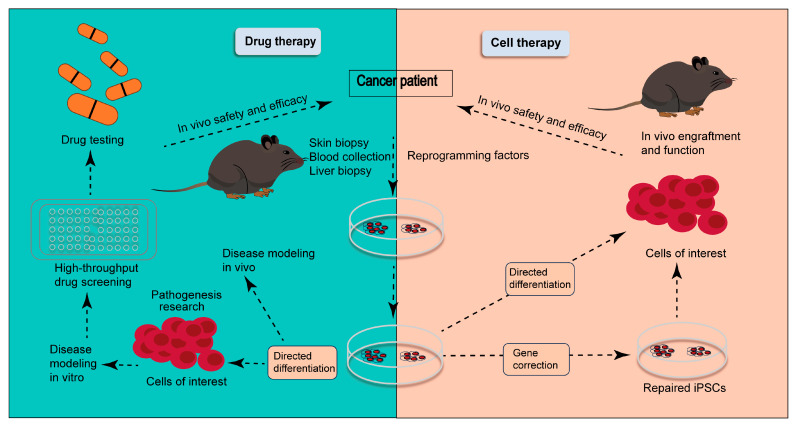
Application of iPSCs in clinical cancer treatment. The application strategies of iPSCs in clinical cancer treatment are mainly divided into two categories: cellular therapy and pharmaceutical therapy. The cellular approach capitalizes on the regenerative potential of iPSCs, while the pharmaceutical application harnesses their modeling capacity for targeted drug development.

**Figure 2 biomedicines-13-02012-f002:**
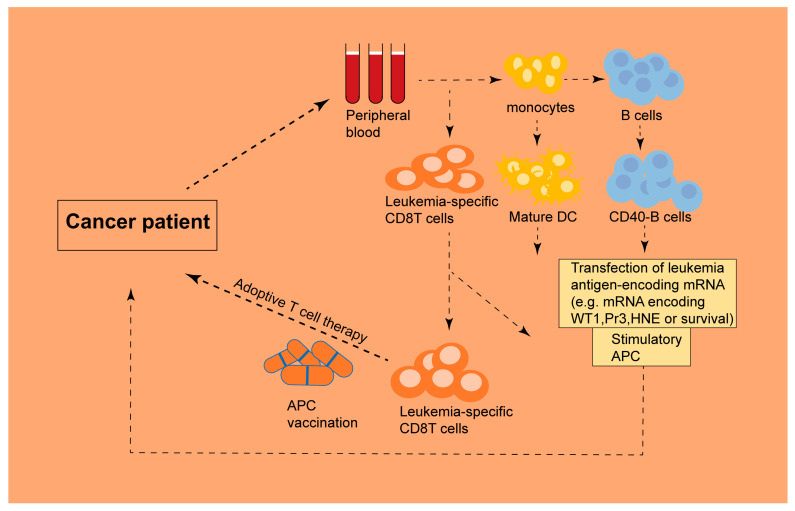
Ex vivo expansion of immune cells from human peripheral blood for antitumor immunotherapy. Human peripheral blood serves as a viable source for the ex vivo expansion of diverse immune cell populations. These expanded immune cells can be functionally engineered to present tumor-associated antigens through antigen-loading protocols. Subsequently, the antigen-primed immune cells are reinfused into the patient via adoptive cell transfer, where they mediate targeted cytotoxicity against malignant cells.

**Table 1 biomedicines-13-02012-t001:** Comparison of ESCs vs. iPSCs.

Parameter	Embryonic Stem Cells (ESCs)	Induced Pluripotent Stem Cells (iPSCs)
Source	Derived from the inner cell mass of blastocyst-stage embryos	Generated by reprogramming somatic cells (e.g., skin or blood cells) using transcription factors (e.g., Oct4, Sox2, Klf4, c-Myc)
Ethical Concerns	High (requires destruction of embryos)	Low (no embryo destruction required)
Immune Rejection Risk	High (allogeneic, may require immunosuppression)	Low (autologous, patient-specific cells reduce rejection risk)
Tumorigenicity	High (risk of teratoma formation)	Moderate (depends on reprogramming method; viral vectors may increase risk)
Reprogramming Needs	Naturally pluripotent (no reprogramming needed)	Requires artificial reprogramming (may introduce genetic abnormalities)
Epigenetic Memory	None (fully reset epigenetic state)	May retain epigenetic marks from the original cell type, affecting differentiation
Clinical Applications	Limited due to ethical and immune concerns	Broader potential (disease modeling, personalized medicine, drug screening)
Future Prospects	Declining use due to ethical issues; mainly used in research	Promising for regenerative medicine, gene therapy, and organoid development

**Table 2 biomedicines-13-02012-t002:** Comparison of different methods used for iPSCs generation.

Method Name	Vector Type	Integration Potential	Efficiency	Safety	Advantages	Disadvantages
Retroviral	Retrovirus	High (integrates into host genome)	Moderate (~0.1%)	Low (risk of insertional mutagenesis and oncogene activation)	First successful method (Yamanaka factors) Stable long-term expression	High tumorigenicity risk Limited clinical applicability
Adenoviral	Adenovirus	None (episomal)	Low (~0.001%)	High (no genomic integration)	No genomic integration Lower tumor risk	Low efficiency Transient expression
Sendai Virus	RNA virus (Sendai)	None (cytoplasmic replication)	High (~1%)	Moderate (viral clearance required)	High efficiency No genomicintegration—works with difficult cell types	Requires rigorous viral clearance Potential immunogenicity
Episomal Plasmids	Plasmid DNA	Low (episomal, transient)	Low (~0.01%)	High (no integration)	Simple and cost-effective No viral components	Low efficiency Requires multiple transfections
mRNA Reprogramming	Synthetic mRNA	None	Moderate (~0.1–1%)	High (non-integrating)	High safety profile No genetic modification—scalable	Technically challenging Requires repeated delivery
Protein Transduction	Recombinant proteins	None	Very low (~0.001%)	High (no genetic material)	No genetic manipulation Ideal for clinical use	Extremely low efficiency Complex purification
PiggyBac Transposon	Transposon	High (integrates but is removable)	Moderate (~0.1%)	Moderate (excision required)	Reversible integration Higher efficiency than plasmids	Excision may leave genomic scars Residual integration risk
CRISPR Activation	CRISPR-dCas9 (activation)	None(epigenetic)	Low (~0.01%)	High (no DNA modification)	Precise epigenetic reprogramming No exogenous genes	Low efficiency Off-target effects possible

**Table 3 biomedicines-13-02012-t003:** The various immunocytes differentiated from iPSCs that are currently being utilized in cancer therapy.

Cell Type	Differentiation Protocol	Therapeutic Applications	Clinical Status	Key Advantages/Challenges	References
iPSCs-NK Cells	Feeder-free/serum-free protocols with IL-15/IL-2/FLT3L CAR engineering (e.g., NKG2D-2B4-CD3ζ)	Targeting hematologic malignancies (AML, lymphoma) and solid tumors (ovarian, breast) Enhanced ADCC via CD16 engineering	Phase I/II: FT522 (Fate Therapeutics), CNTY-101 (Century Therapeutics)	Advantages: Off-the-shelf, low GVHD risk.	[26]
iPSCs-T Cells	Stroma-free platforms with Notch signaling G9a/GLP inhibition to enhance maturation CAR-T engineering (e.g., CD19, CD70)	Hematologic cancers (B cell malignancies, AML) Solid tumors (renal cell carcinoma)	Preclinical/Phase I: FT819 (CAR-T for SLE, potential cancer crossover)	Advantages: Unlimited supply, rejuvenated telomeres Challenges: Immature phenotype	[27]
iPSCs-DCs	GM-CSF/IL-4 differentiation from hematopoietic progenitors	Antigen presentation, tumor vaccine platforms	Preclinical: Limited trials due to manufacturing complexity	Advantages: High antigen uptake Challenges: Low migratory capacity	[28]
iPSCs-CAR-NKT	IL-15/IL-21-driven differentiation CAR engineering (e.g., CD19, CD70)	Dual targeting of tumors and immunosuppressive TME	Phase I: NKT cell trials for head/neck cancers (Japan)	Advantages: Innate/adaptive immunity synergy Challenges: Scalability	[29]
iPSCs-Macrophages	M-CSF/IL-3 differentiation CAR-M engineering (e.g., anti-CD19)	Solid tumor infiltration (e.g., glioblastoma)	Preclinical: Emerging interest	Advantages: TME remodeling Challenges: Pro-tumor polarization risks	[28]

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
