# Peer review of "Induced Pluripotent Stem Cell-Based Cancer Immunotherapy: Strategies and Perspectives"

_biomedicines, 2025, doi:10.3390/biomedicines13082012_

Round 1
Reviewer 1 Report
Comments and Suggestions for Authors
I found this manuscript to be a thorough and timely review of iPSC-based cancer immunotherapies. It covers a wide range of topics, from protocols for differentiating iPSCs into various immune cells to innovative strategies like CAR and CRISPR engineering, and it highlights important challenges and future directions. This topic is highly relevant and could make a valuable contribution to the field.
However, in its current form, the manuscript is much too long and reads more like an exhaustive technical manual than a focused review. There are many sections where the same information is repeated in detail, which makes it difficult to follow. For instance, the protocols for differentiating iPSCs into NK cells, T cells, macrophages, dendritic cells, and B cells all include very similar descriptions of embryoid body formation and OP9 coculture systems. These could easily be consolidated into a single section or summarized in a table. Additionally, the discussions on CAR engineering and strategies for overcoming the tumor microenvironment are revisited multiple times with similar wording, and the advantages of iPSCs over ESCs are restated repeatedly in the introduction.
I recommend the authors:
-
Streamline the repetitive content so that key points are covered clearly without unnecessary repetition.
-
Provide a more critical analysis of which strategies appear most promising or have faced significant challenges, and clearly outline research priorities for moving the field forward.
-
Update the review with recent clinical trial data to enhance its relevance and strengthen the discussion of clinical translation.
-
Add more summary tables or schematic figures—right now, there are only two figures, which isn’t enough for a review of this scope. Moreover, the current figures look low in resolution and should be replaced with higher-quality versions suitable for publication.
I recommend that the authors carefully check the English language throughout the manuscript. Some sections, especially the detailed protocol descriptions, contain long and awkwardly structured sentences that reduce clarity and readability. Apart from that, please correct minor language issues, such as:
-
On approximately pages 3–4, “ESCs requires” should be “ESCs require.”
-
Also on pages 3–4, “similar the patterns” should be corrected to “similar patterns.”
-
On pages 5–6, the phrase “early phage clinical trials” seems to be a typo for “early phase clinical trials.”
Reviewer 2 Report
Comments and Suggestions for Authors
The manuscript by Xiaodong Xun et. al. presents a comprehensive overview of the role of iPSC-derived immunocytes in cancer immunotherapy, detailing the differentiation of iPSCs into various immune cell types and their prospective applications in combating malignancies. Moreover, the inclusion of cutting-edge strategies, including CAR engineering, CRISPR-based genome editing, nanoparticle-assisted differentiation, and 3D culture systems, makes this review a significant addition to this field of research.
The only thing I find lacking is more visual summaries, such as comparison tables or decision flowcharts, which could enhance accessibility and make the article more balanced. Overall, this is a well-written, forward-looking review that makes a significant contribution to the understanding of iPSC-based immunotherapy and provides a strong foundation to translate stem cell technologies into clinical cancer treatment.
Reviewer 3 Report
Comments and Suggestions for Authors
The manuscript is well-written and discusses an important topic, highlighting the role of iPSC-derived immunocytes in cancer therapy. It presents current applications clearly and follows a well-organized structure. The review effectively emphasizes the therapeutic potential of NK cells, CAR-T cells, and others. However, a few key methodological and foundational aspects are missing. Addressing these points will further strengthen the paper before publication.
The sentence “Most chemotherapy agents have no effect on these patients after some courses of treatment, and their malignancy ultimately can recur” is overly broad and clinically inaccurate. The sentence should be like “Most chemotherapy agents gradually lose effectiveness in these patients after several treatment cycles, leading to eventual recurrence of the malignancy.”
The manuscript skips a focused section on iPSC Reprogramming and Differentiation Techniques. The author needs to include a comprehensive section covering: Reprogramming somatic cells into iPSCs (mechanisms, vectors, and protocols). Quality control parameters for selecting functional iPSC-derived immunocytes and Differentiation of iPSCs into immunocytes like T cells, NK cells, DCs, B cells.
The manuscript does not distinguish between these two key NK cell (Autologous vs. Allogeneic NK Cells) sources. The author needs to add a comparative discussion of: Autologous NK cells: Derived from the patient, reduced immune rejection but may have compromised functionality. Allogeneic NK cells: From donors or iPSC banks, higher activity but risk of immune mismatch or graft-versus-host disease.
The review moves directly into applications without establishing a biological foundation Explanation of iPSC Biology. The should include a section briefly explaining; the biological characteristics of iPSCs, Key transcription factors involved in reprogramming (e.g., Oct4, Sox2, Klf4, c-Myc) and Pluripotency and self-renewal capabilities.
No structured comparison is presented in this review. Insert a Table titled “Comparison of ESCs vs. iPSCs” including the parameters: Source, ethical concerns, immune rejection risk, tumorigenicity, reprogramming needs, epigenetic memory, clinical applications, and future prospects.
The diversity in iPSC reprogramming strategies is not discussed. Insert a Table titled “Comparison of Different Methods Used for iPSC Generation” including: Method name, vector type, integration potential, efficiency, safety, advantages and disadvantages. This table will help readers, especially researchers and clinicians, choose the most appropriate reprogramming approach for specific applications.
Round 2
Reviewer 1 Report
Comments and Suggestions for Authors
While the authors have addressed the scientific content adequately, Figures 1 and 2 remain suboptimal in quality. The resolution is still low, and the cartoons are visually unappealing and overly simplistic. For a comprehensive review of this scope, high-quality schematics that are both informative and visually engaging are important. The authors are strongly encouraged to professionally redesign these figures to improve clarity and publication readiness.
